# A Miniaturized Microbe-Silicon-Chip Based on Bioluminescent Engineered *Escherichia coli* for the Evaluation of Water Quality and Safety

**DOI:** 10.3390/ijerph18147580

**Published:** 2021-07-16

**Authors:** Emanuele Luigi Sciuto, Domenico Corso, Sebania Libertino, Jan Roelof van der Meer, Giuseppina Faro, Maria Anna Coniglio

**Affiliations:** 1Azienda Ospedaliero Universitaria Policlinico “G. Rodolico-San Marco”, Via S. Sofia 78, 95123 Catania, Italy; e.l.sciuto@gmail.com; 2Istituto per la Microelettronica e Microsistemi–Consiglio Nazionale delle Ricerche (CNR-IMM), Ottava Strada 5, 95121 Catania, Italy; 3Department of Fundamental Microbiology, Bâtiment Biophore, University of Lausanne, 1015 Lausanne, Switzerland; janroelof.vandermeer@unil.ch; 4Azienda Sanitaria Provinciale di Catania, Via S. Maria La Grande 5, 95124 Catania, Italy; giuseppina.faro@aspct.it; 5Regional Reference Laboratory of Clinical and Environmental Surveillance of Legionellosis, Catania, Department of Medical and Surgical Sciences and Advanced Technologies “G.F. Ingrassia”, University of Catania, Via Sofia 87, 95123 Catania, Italy

**Keywords:** water analysis, engineered *Escherichia coli*, miniaturizable optical system, mercury sensing, SiPM

## Abstract

Conventional high throughput methods assaying the chemical state of water and the risk of heavy metal accumulation share common constraints of long and expensive analytical procedures and dedicated laboratories due to the typical bulky instrumentation. To overcome these limitations, a miniaturized optical system for the detection and quantification of inorganic mercury (Hg^2+^) in water was developed. Combining the bioactivity of a light-emitting mercury-specific engineered *Escherichia coli*—used as sensing element—with the optical performance of small size and inexpensive Silicon Photomultiplier (SiPM)—used as detector—the system is able to detect mercury in low volumes of water down to the concentration of 1 µg L^−1^, which is the tolerance value indicated by the World Health Organization (WHO), providing a highly sensitive and miniaturized tool for in situ water quality analysis.

## 1. Introduction

The chemical state of water intended for human usage (drinking, domestic, and recreational in general) needs a thorough and constant analysis in order to prevent any negative consequences for public health. Chemical pollutants such as pesticides, salts, toxins, drugs, and metals are, in fact, frequently released in ground and surface water becoming a cause of concern for population safety and preservation of natural resources [1,2].

Water characteristics may affect the distribution system and vice versa. Very soft water [3], high concentrations of soluble sulfates, chlorides, and natural organic matter [4], as well as dissolved oxygen [5] can dramatically affect pipe corrosion. On the other hand, the presence of metals in water can be due to the quality and material of pipelines, as well as their operating conditions. Finally, at low pH conditions, carbon dioxide forms carbonic acid causing corrosion when combined with iron [6].

Mercury is a common plumbing contaminant that can be found in elemental/metallic (Hg°), organic and inorganic forms, such as the divalent inorganic mercury (Hg^2+^). It can be discharged with wastewater into water systems from many different sources: spills from devices (e.g., broken thermometers, flowmeters, etc.); laboratory reagents; amalgam from dental clinics; as a contaminant in some types of janitorial chemicals; or released by industrial activities [7,8]. Hg^2+^ in water poses severe risks for the population because small amounts can cause extensive plumbing contamination. It may accumulate in joints and low water flow areas of plumbing systems where it may persist for many years, or it can dissolve and combine with other metals (e.g., copper), forming liquid or solid amalgams [9]. Hg^2+^ can accumulate in biofilms on the internal surfaces of pipes and may be transformed into other more toxic forms by bacterial activity. In particular, sulfate-reducing bacteria such as *Desulfobacteraceae* and *Desulfovibrionaceae*, which are resistant to the toxicity of inorganic mercury compounds, metabolize, and convert the inorganic mercury (Hg^2+^) into methylated organic compounds, which are highly neurotoxic to animals, persist and bioaccumulate [10]. Finally, Hg^2+^ contaminated pipelines may become a source of antibiotic-resistant bacteria due to the co-selection of mercury and antibiotic resistance gene pools [11].

The tolerance limit for Hg^2+^ in water for public health safety has been defined by the World Health Organization (WHO) at 1 μgL^−1^ [12]. At higher concentrations and depending on the ingestion volumes and routes, there is an increased risk for the development of neurological disorders, skin rash, or kidney failure [13]. In this scenario, integrated response strategies to deal with any possible threat from Hg^2+^ concentrations above the tolerance limit need to be developed. This approach is termed *water safety plan* (WSP) and comprises proactive risk management methods along with analytical technologies. A WSP can vary in complexity because it draws on the multiple-barrier approach, which recognizes that while each individual barrier may not be able to completely remove or prevent contamination and, therefore, protect public health, the barriers work together to provide water safety. To this aim, in situ monitoring can help to record the spatial and time variation of Hg^2+^ in water, as well as to identify contamination events early enough to allow an effective response.

Current analytical approaches can provide high-performance detection and quantification of chemical traces in water samples. However, most of these approaches are not well suitable for in situ monitoring because they need bulky and costly laboratory instruments, and high-cost operational and sophisticated procedures, requiring specialized personnel.

Chromatographic and spectrometric methods [14,15,16,17] are the gold standard for most of heavy metal detection in water. This includes the quantification of mercury. They are able to operate at high throughput. In contrast, they also necessitate long measurement time, requiring mercury reduction to the elemental state; pre-concentration steps [18,19,20]; expensive reagents; big instrumentations, and highly trained personnel [21,22]. Hence the need for simpler and smaller instruments, including alternative detection methods.

Many efforts, in this sense, have been spent trying to integrate and miniaturize the entire analytics, providing small portable instruments without affecting the quality of analysis. For example, low-cost colorimetric and fluorometric methods have been incorporated into portable platforms, by which metal presence is detected through interaction with specific dyes and subsequent color or light changes. However, colorimetric strategies are, often limited by the risk of cross-reactivity with unspecific metals (e.g., zinc and cadmium compared to mercury), causing false positive detection [23,24,25]. Fluorometric methods frequently lack robustness due to the conventional dyes being applied (e.g., cyanine, FAM, etc.) that, typically, degenerate after prolonged usage and affect the sensitivity of the analysis [26]. Thus, reliable, inexpensive, and in situ surveillance systems for metal detection are still needed.

Silicon-based technology provided an important breakthrough in environmental analysis. Silicon is a biocompatible substrate that allows embedding the whole analytical chain. It is suitable for a wide series of implementations such as detector downscaling (integrated microelectrodes, micro-sized photomultipliers, etc.) [27,28,29] and integration of microfluidics [30], which make it perfectly suitable for in situ measurements.

In particular, about 20 years ago, a very sensitive class of Si-based photodetectors was proposed [31]: the silicon photomultipliers (SiPM). SiPM is formed by an array of single-photon detectors, each one operating in Geiger mode above its breakdown voltage and all connected to a common load [32,33,34,35]. When the detector is hit by a photon, it experiences an avalanche breakdown providing a strong amplification (at least a million times) of the incoming signal. The final electrical current is the sum of the avalanche currents coming from the hit cells on the common load; hence, it is proportional to the number of photons hitting the device. SiPM is optimally suited for low light illumination but becomes “blind” if the photon rate is too high. It should be mentioned that when illumination is high, the device does not break (as it occurs to traditional photomultiplier tubes), it is only blind to a photon rate above a threshold provided by the time needed for the device to return in its quiescent state after the breakdown.

In parallel, other technologies called whole-cell biosensors received widespread attention for their innovative properties. These sensing systems are based on the bioactivity of microorganisms, especially bacteria, which were genetically modified in order to be sensitive towards specific analytes [36,37]. Whole-cell biosensors exist for a variety of heavy metals. They produce specific optical, colorimetric, and/or electrochemical signals in response to the interaction of the living cell with the target molecule. The engineered microorganisms are physiologically more robust than conventional probes and dyes in case of prolonged usage and allow a high degree of portability due to the survival at different environmental conditions [38]. Moreover, their specific interactions with the target enhance the selectivity avoiding any case of cross-reactivity [39].

Here, we propose an advanced and miniaturized method for the specific and highly sensitive detection and quantification of divalent mercury in water combining the properties of both whole-cell biosensor and silicon-based technology. The analytical system uses engineered *Escherichia coli* that produces bioluminescence in response to mercury in a dose-dependent manner. Bioluminescence is detected by the SiPM, allowing quantification of the target. We tested the performance of the system as a function of mercury concentrations and examined its level of portability.

## 2. Materials and Methods

### 2.1. Materials

Dilutions were made using a filtered sterile 10X phosphate-buffered saline solution or PBS that was prepared by mixing 137 mM NaCl, 2.7 mM KCl, 4.3 mM Na_2_HPO_4_, and 1.47 mM KH_2_PO_4_ (Thermo Fisher Scientific, Waltham, MA, USA) at pH 7. Bacteria cultures were prepared using sterile Luria Bertani (LB) broth or Buffered Peptone Water (Thermo Fisher Scientific, Waltham, MA USA). Mercury standards were prepared by dissolving Mercury(II) Chloride (HgCl_2_) powder (Sigma-Aldrich, St. Louis, MO, USA). For the optical analysis, 12.5 mm × 12.5 mm × 45 mm disposable UV-cuvettes (BRAND GMBH, city, Germany) and 4 mL glass vials (14.7 mm × 45 mm) (Lab Logistics Group International GmbH, Meckenheim, Germany) were used. The SiPM consisted of 6.07 × 6.07 mm^2^ (MICROFJ-60035) with 22,292 cells and was purchased from ON Semiconductors.

### 2.2. Sensing Element

The sensing strain used for the selective recognition of Hg^2+^ in water was *Escherichia coli* DH5α that was transformed with the recombinant plasmid pSB403-merR_luxCDABE, as reported in reference [36]. The plasmid contained a fusion between the gene *merR,* coding for the transcription repressor of the mercury resistance operon, and the structural genes *luxCDABE*, encoding the bacterial luciferase (Lux) and its long-chain aldehyde substrate, all regulated from the *PmerT* promoter. Thus structured, the *E. coli* cells induced the expression of *lux* genes once they internalized the Hg^2+^ from the water sample, as schematized in Figure 1. The analyte binds the MerR protein in the cell (red in the figure), causing it to release its repression on the promoter, which triggered the expression of the luciferase and its substrate (blue in the figure). The enzyme then catalyzed the long-chain aldehyde oxidation to a carboxylic acid and released energy as photons [40]. This gave a 490 nm blue-green light, used as output for the optical detection and quantification, whose intensity was related to the mercury amount in the water sample in a dose-dependent manner.

The sensor assay was operated by suspending the *E. coli* cells in 10 mL of tetracycline-supplemented (10 µg mL^–1^) Luria Broth (Sigma-Aldrich, St. Louis, MO, USA) medium and incubating this culture for 12–16 h at 37 °C and 150 rpm. The culture was 50-fold diluted the next day in the fresh medium of the same and grown to an optical density (OD_600_) of 0.6, indicative of the exponential phase of growth. The culture was then 10-fold diluted in peptone water minimal medium and used to prepare the analytical samples.

For the optical signal evolution analysis, the *E. coli* culture was grown up to an OD_600_ of 0.1. This OD value exemplified early logarithmic growth, during which the cells were the most physiologically active. Then, incubation at room temperature was observed over time as an increase of the emitted light.

### 2.3. Optical System

The optical system used to monitor the Hg^2+^ presence in water was schematically shown in Figure 2.

It was mainly composed by a SiPM as photodetector; a Source Meter Unit (Keithley 2636B) to bias the photodetector and read the electro-optical signal from the device; an isolated dark box to avoid any external light, maintain temperature, and humidity (data not shown); each acquisition chamber was optically insulated, using a black tape; and a properly developed software to control signal measurement and acquisition times.

The SiPM was a solid-state photodetector composed of an array of pixels. Each pixel was a single-photon avalanche diode (SPAD), operating in digital mode. It provided its maximum current when a photon generated an electron-hole pair and no current when it was in a quiescent state (Geiger mode) [35,41,42,43]. To have an analog output, i.e., a current proportional to the impinging photons number, all pixels were connected in parallel to a common resistor of some 10s of Ω, where the output signal was collected. The breakdown voltage (BV) was the minimum reverse bias at which each SPAD operates in Geiger mode, while the overvoltage (OV) was the excess voltage applied above BV. Both BV and OV ranges were a function of the particular SiPM technology.

In the following experiments, we used 2 different technologies to define the best for our application. The system sensitivity was a function of the device’s dark current during operation, and currents of 80 pA and 6 nA were measured for the 2 technologies with a measurement uncertainty below 10% in the worst case.

Current-voltage measurements in the range from −23 V to −32 V were acquired in dark and low light illumination conditions and the best operating parameters in terms of higher gain and lower noise were experimentally defined. In this range of measurement, we found the BV at −24.4 V. Therefore, 2 OV were chosen in order to maximize the system response under illumination: −24.7 V, and −25.4 V. The 1st value (−24.7 V) was chosen in order to reduce as much as possible the dark current (i.e., the current value measured in dark) and allow small optical signals to be measured above the signal-to-noise-ratio (SNR). It was used to determine the device sensitivity and corresponds to an OV of 0.3 V with an amplification factor above 10^5^. The 2nd value (−25.4 V) was chosen in order to have the highest gain (above 10^6^).

The device, mounted on a board as described in reference [44], was placed within a dark box with a square cuvette containing the sample to test on top (Figure 2a). The cuvette dimensions were roughly the same as the SiPM device active area in order to maximize the collection region. A photograph of the device is reported in Figure 2b. The electrical signal is transferred to and recorded by a PC, and the measurement system was controlled by a custom-developed program written using the Labview^®^ software (see schematics in Figure 2c). Acquisition times ranged from 100 s to 24 h, with step points selectable from a few seconds to hours, depending on the measurement.

Finally, all the acquired signals were elaborated offline by a routine developed in MATLAB™, version 2017a.

### 2.4. Bioluminescence Analysis

The bioluminescence analysis was performed on water samples contaminated by known concentrations of Hg^2+^. These were prepared by serial dilutions of a 25 mM HgCl_2_ stock solution in sterile tap water to 0.25; 1; 2.5; 10; 25; 35; 50; 100 and 200 μgL^−1^. A total of 200 µL of Hg^2+^ standards solutions (the same volume of tap water was used as reference) were mixed to 1.8 mL of the *E. coli* culture, prepared as described above (see Section 2.1), and incubated at 37 °C and 350 rpm for 1, 2, and 3 h in order to accelerate the mercury uptake process by *E. coli* and the subsequent luciferase expression. The optimization of growth conditions is important since such factors as the bacteria energy state, the lux synthesis/degradation rate and the amount of incoming/recycled mercury can influence the optical performances of luciferase by altering the lux bioluminescence quantum yield from its normal range of 0.1–0.16 [45,46,47,48,49,50].

Then, 200 µL of induced mixes were transferred in the cuvette of the optical system, already positioned on top of the SiPM (see Section 2.2). The dark box was then sealed, and the measurement was started.

To analyze the development of the optical signal over time, samples were prepared by mixing the 25 μg L^−1^ Hg^2+^ solution with the *E. coli* culture at OD_600_ of 0.1 and, then, spotted in the cuvette without the incubation step and directly observed at room temperature.

The acquisition times were varied in order to obtain information on the luminescence evolution as a function of time, ranging from 100 s, in the first set of experiments, to 2, 3 h or days for analysis of time development.

## 3. Results

In order to determine the dose-response of the bioluminescence output to varying Hg^2+^ concentrations in the sample, *E. coli* cultures with an OD_600_ of ~ 0.6 were mixed to mercury standard solutions ranging from 0.25 to 200 µg L^−1^. Electro-optical signals acquired during 100 s clearly increased as a function of Hg^2+^ input (Figure 3). Signals slightly decreased during the 100 s period but were stable for the first 10 measurement points. A concentration of 1 µg L^−1^ Hg^2+^ was easily differentiable from the reference signal after a 2 h incubation time (Figure 3). A slight signal saturation occurs over 100 µg L^−1^, indicating that a quantitative Hg^2+^ measurement in the range from 1 to 200 µg L^−1^ is feasible without any further sample dilution.

In order to examine the effect of prolonged incubation time on the optical response of the assay, we averaged the signal currents from the first 10 measurements points (at each concentration, as in Figure 3) for incubations of 1 h up to 3 h (Figure 4, colored lines). This showed that incubation time is more relevant for higher concentrations; from 10 µg L^−1^ a significant increase in the current value was observed as a function of the incubation time. On the contrary, at lower Hg^2+^ concentrations, the trend may actually be inverse, and longer incubation times tend to lower the bioluminescence output (Figure 4, 1 µg L^–1^). An incubation time of 1 h is, therefore, sufficient for measuring Hg^2+^ concentrations above 1 µg L^–1^.

It is not possible to infer the essay behavior below 1 µg L^–1^ since we were too close to our sensitivity limit. However, Berset et al. in 2017 [51] developed a mechanistic model of an arsenic reporter construct, quite similar to that used in our mercury sensitive *E. coli*, by which it is possible to better understand the intracellular biochemistry involved in the arsenic and, in general, the heavy metals sensing in water. The model allowed, for example, to find that changes of specific parameters such as the ArsR/DNA and the ArsR-dimers/arsenite binding affinity affect the cell’s response in a different way depending on the arsenic concentrations range they are exposed.

Because of the tendency of recorded signals to slightly decrease over time (Figure 3), we looked more closely at the development of the bioluminescence signal over time in samples at room temperature directly incubated on the SiPM cuvette. To perform such measurements, the optical set-up was entirely re-designed by adding a second detection site (inset of Figure 5), i.e., a second SiPM and a cuvette with the water sample and the whole electronics and software to address and control it. The aim of the second detection site was twofold. It was used either to monitor in real-time both the treated and the control sample or to monitor a replicate sample. Figure 5 shows the replicate bioluminescence signal as a function of the incubation time at room temperature on the chip from a mixed *E. coli* culture at 0.1 OD_600_ with 25 µg L^−1^ Hg^2+^.

These results indicated that the bacteria survived in the detection chamber for the entire measurement time and produced detectable output even at room temperature. The signal started to rise above the background noise after roughly 20 min (10^3^ s), with a maximum observed after 2 h (Figure 5). This indicated that incubation at room temperature is a viable method for Hg^2+^ detection by *E. coli* whole-cell sensors.

The delay between the first contact with Hg^2+^ and emission of the bioluminescence signal by the bacteria is caused by the time needed for de novo transcription and translation in the cells of the reporter gene constructs, and the time for all cells to react—given that there is some cell-to-cell variability in response. The observed signal is the integral of the signal emitted by many individual bacterial cells, and only when this surpasses the background noise, it can be measured by the SiPM detectors. Biochemically speaking, the cells “recycle” the Hg^2+^ and cannot transform it; therefore, the Hg^2+^ will continue to derepress the MerR repressor, leading to an accumulating signal over time until the cells run out of energy.

In order to understand how long the cells can continue to produce a response, we prepared from the same *E. coli* culture five dilutions and stored them at 5 °C. One dilution was taken every 3–4 days, incubated overnight at 37 °C and 150 rpm up to OD_600_ 0.1, and mixed with a 25 μg L^−1^ Hg^2+^ solution. Time measurements in Figure 6 indicated that the onset of the detectable response was almost the same for all storage times (i.e., around 20–30 min). In contrast, the absolute signal decreased by 25–30% for every 3 days longer storage (Figure 6), which was probably the consequence of a loss of cell viability.

## 4. Conclusions

Maintaining the safety of drinking water is a challenge for the Public Health sector since water distribution systems are complex networks and drinking water quality may be affected by many factors. Accumulation and biotransformation of Hg^2+^ in plumbing systems can threaten public health because small amounts can cause extensive plumbing contamination. As already described elsewhere [52,53], for a comprehensive risk assessment, all the system’s vulnerability needs to be assessed through a water safety plan (WSP). Tracking concentrations of Hg^2+^ in a water system requires the ability to monitor its concentration in situ. New technologies have been developed to detect metals in water without requiring the collection and preparation of the water sample. The miniaturized system described in this paper allows the easy and rapid quantification of Hg^2+^ levels in water samples. The system is based on the synergy between a bioluminescent *E. coli*—used as a sensing element for the selective recognition of Hg^2+^—and a sensitive micro-sized Silicon Photomultiplier, used as a detector for the bioluminescence. The system can be fully integrated in a dark box of a few cubic centimeters and further parallelized to contain multiple detectors.

Our results show reliable detection of Hg^2+^ at 1 µg L^–1^, which is the tolerance value indicated by the WHO, within 2 h incubation time.

The system can be further simplified and optimized to allow analyte detection directly on chip without separate preincubation, and our data suggest that 30 min to a few hours are sufficient for a correct measurement at room temperature. *E. coli* cultures can be sustained for a week at 5 °C and still produce sufficient output, although cold storage decreases the performance of the cells.

Given the water system’s complexity and the unstable water quality conditions, prediction and monitoring of Hg^2+^ within a distribution system is challenging. For this reason, an integrated approach is needed for effective and efficient water quality management. This approach will comprise technical tools along with proactive risk management methods such as a WSP. The reported properties, together with the high degree of miniaturization and integration, make the optical system described in the paper suitable for in-situ applications of water quality control.

## Figures and Tables

**Figure 1 ijerph-18-07580-f001:**
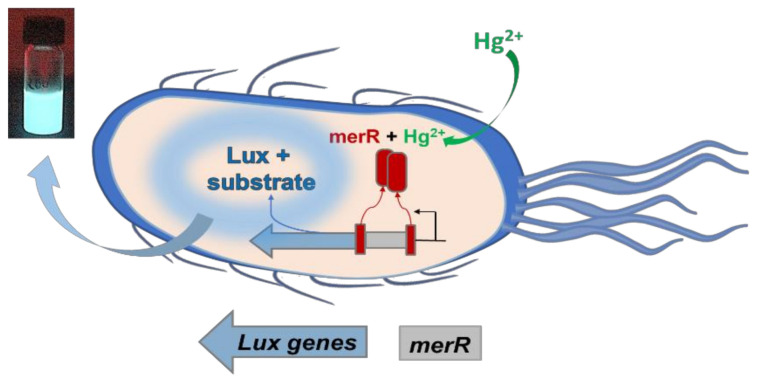
Scheme of the Hg^2+^ recognition mechanism performed by the engineered *E. coli* sensor cells.

**Figure 2 ijerph-18-07580-f002:**
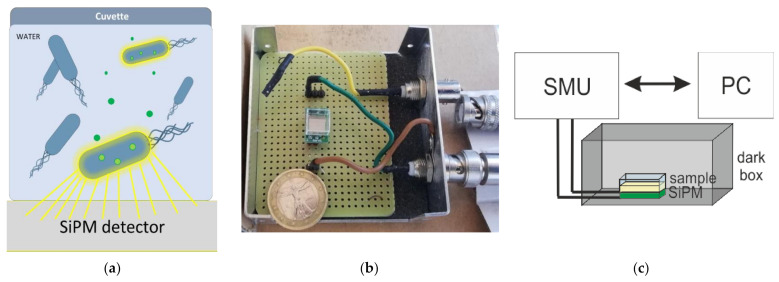
(**a**) schematic of the optical system operation principle based on the Hg^2+^ (green circle) uptake by bioluminescent *E. coli* (blue); (**b**) photograph of the detection site; (**c**) schematic of the optical system configuration, where the SMU is the source meter unit.

**Figure 3 ijerph-18-07580-f003:**
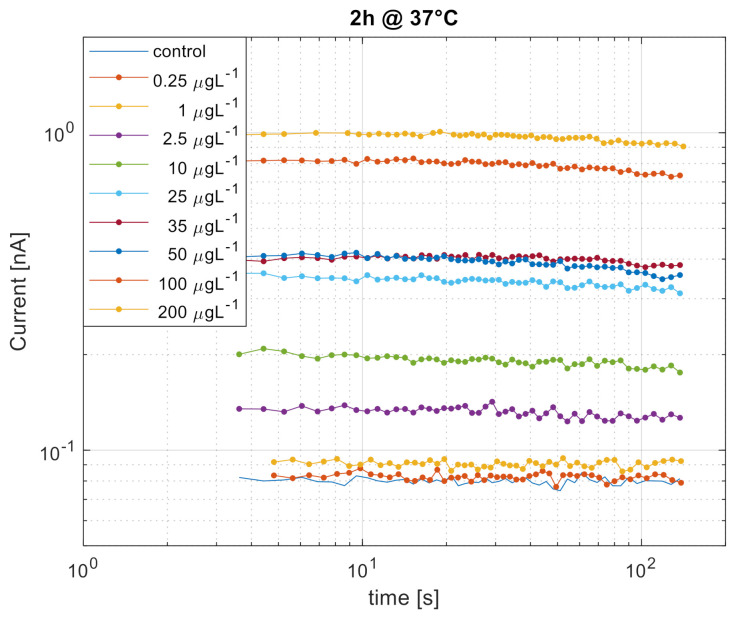
Mercury induced bioluminescence from the *E. coli* whole-cell biosensors detected by the SiPM instrument. Colors correspond to concentrations as explained in the legend inset. Samples were incubated with *E. coli* at 37 °C for 2 h, after which the signal was acquired for about 100 s.

**Figure 4 ijerph-18-07580-f004:**
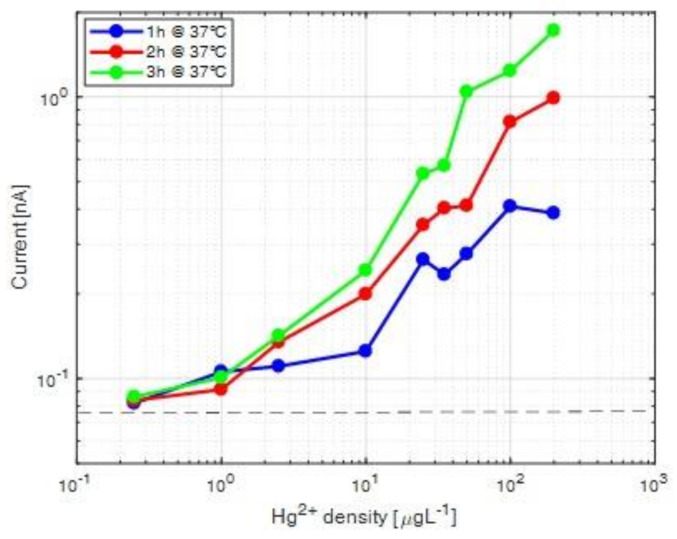
Average bioluminescence signals as a function of Hg^2+^ concentration from whole-cell sensor incubations at 1 h (blue line and symbols), 2 h (red line and symbols) or 3 h (green line and symbols). Each point is an average of the first 10 measurement points by the SiPM (as in Figure 3). The Black dashed line corresponds to the dark current.

**Figure 5 ijerph-18-07580-f005:**
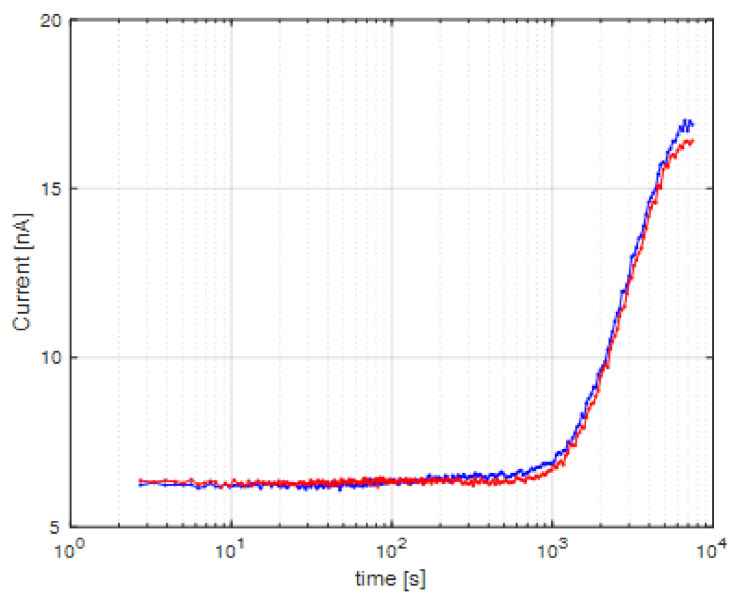
Simultaneous bioluminescence signal measurement from two replicate samples as a function of incubation time on SiPM chip. Solutions contained *E. coli* with OD_600_ of 0.1 mixed with 25 µg L^−1^ of Hg^2+^, incubated at room temperature. Inset: re-designed optical set-up containing two detection sites.

**Figure 6 ijerph-18-07580-f006:**
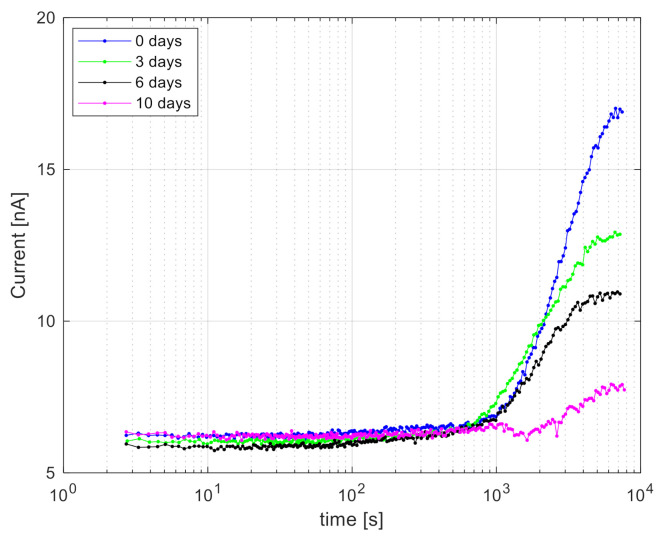
Bioluminescence signal measured as a function of cell culture storage time. Diluted *E. coli* suspensions were stored for 0, 3, 6, and 10 days at 5 °C, reactivated at 37 °C for 12 h to an OD_600_ at 0.1, mixed with 25 µg L^–1^ Hg^2+^, and directly measured at room temperature.

## Data Availability

Not applicable.

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
