# Peer review of "A Miniaturized Microbe-Silicon-Chip Based on Bioluminescent Engineered Escherichia coli for the Evaluation of Water Quality and Safety"

_ijerph, 2021, doi:10.3390/ijerph18147580_

Round 1

Reviewer 1 Report

The topic of the manuscript is using engineered Escherichia coli bacteria for quantitative detection of mercury (Hg) in water. The bacteria was modified to produce light when mercury ions are internalized. The light is detected by a multi-pixel silicon avalanche-diode detector, also known as a SiPM.

The manuscript is clearly written and interesting to a substantial readership.

There are some minor issues in the manuscript that could be addressed.

First, I would recommend against using the word 'cheap', as it has acquired a meaning of being of low quality. 'Inexpensive' would be a better choice here.

Second, when specifying current in Figs. 3-6, consider using nano Amperes [nA] or pico Amperes [pA], or corresponding multipliers (x10^-9, or x10^-12).

Finally, I would also recommend providing an electrical schematic of the measurement circuit, most probably in Figure 2.  I am affraid that many readers are not familiar with operation of SiPM detectors.

In conclusion, the manuscript can be published in its present form, while it would be beneficial to introduce changes recommended above.

Author Response

We would like to thank all the referees for the suggestions that help us to improve the quality of our manuscript. In the following we report the referees’ comments/suggestions in bold and our answers below each one in italic. 

First, I would recommend against using the word 'cheap', as it has acquired a meaning of being of low quality. 'Inexpensive' would be a better choice here.

We have substituted the word “cheap” with “inexpensive” in the text

Second, when specifying current in Figs. 3-6, consider using nano Amperes [nA] or pico Amperes [pA], or corresponding multipliers (x10^-9, or x10^-12).

We have changed the figures 3-6 placing in the vertical scale nA as measurement unit, thus avoiding to use the multipliers.

Finally, I would also recommend providing an electrical schematic of the measurement circuit, most probably in Figure 2.  I am afraid that many readers are not familiar with operation of SiPM detectors.

To help readers unfamiliar with this device we added a brief description of the SiPM operation principle in the ‘Material and Methods’ section, with 4 references related to the device characteristics, architecture, operation principle and performances. Moreover, we specified the SiPM is mounted on a PCB provided with the device and added a reference (ref. 45) to the provider data sheet.

Reviewer 2 Report

The authors have performed “A miniaturized microbe-silicon-chip based on bioluminescent engineered Escherichia coli for the evaluation of water quality and safety” that is able to detect mercury in low volumes of water down to the concentration of 1 μgL-1, which is the tolerance value indicated by the World Health Organization (WHO), providing a highly sensitive and miniaturized tool for in situ water quality analysis. Overall, it is a good manuscript, well-written (except for the minor necessity of inclusions to improve the abstract and conclusions, make them supported by data). I can recommend its publication after minor corrections, which are subject to author and editor decision.

  • Page 2: The reference source has been displayed along the text as “Reference source not found”. Could the authors verify the references 19-20, please?
  • Page 5: The authors have analyzed the evolution of optical signal with the coli culture at OD600 of 0.1. Could the authors comment the influence of the Optical density about the luminescence, please?
  • Page 6 (lines 7-8): A concentration of 1 μgL-1 Hg2+ was easily differentiable from the reference signal after 2 h incubation time in bioluminescence measurements. What is the estimate/measurement of the emission efficiency?
  •  Page 6 (lines 14-15): Information on sensitivity and uncertainty values would be quite useful.
  • Page 6 - Figure 3: The bioluminescence using the concentration of 35 μgL-1 Hg2+ and 50 μgL-1 Hg2+ has not been easily differentiable. Could the authors give more information about this issue, please?
  • Page 7 – Figure 4: The average bioluminescence signals have presented unexpected behavior to concentration below 1 μgL-1 Hg2+ with the increase of incubation time. Could the authors give more information about this issue, please?

Author Response

We would like to thank all the referees for the suggestions that help us to improve the quality of our manuscript. In the following we report the referees’ comments/suggestions in bold and our answers below each one in italic. 

  • Page 2: The reference source has been displayed along the text as “Reference source not found”. Could the authors verify the references 19-20, please?

We have corrected the error by removing the cross-references from the manuscript.

  • Page 5: The authors have analyzed the evolution of optical signal with the coli culture at OD600of 0.1. Could the authors comment the influence of the Optical density about the luminescence, please?

Such low OD has no influence on the bioluminescence signal - no particle light scattering expected. Also, this OD exemplifies early logarithmic growth, during which the cells are the most physiologically active. A sentence has been added at page 4 to clarify this point.

  • Page 6 (lines 7-8): A concentration of 1 μgL-1 Hg2+was easily differentiable from the reference signal after 2 h incubation time in bioluminescence measurements. What is the estimate/measurement of the emission efficiency?

We assume the ‘emission efficiency’ required by the referee is the quantum yield of bacterial luciferase (Lux) bioluminescence, i.e., light per energy input, that is known to be in the range 0.1-0.16 (Ref. 46-51 now in the manuscript). The emission efficiency in this case may mean the collected light per emitted light, but we have no absolute way of knowing the absolute emitted light per cell since Lux bioluminescence is function of different factors including the energy state of the cells, the synthesis and degradation rates of the luciferase and newly incoming and recycled mercury on the MerR regulator (referred to our sensing application).

We highlighted this statement on page 6 (Material and methods section).

  • Page 6 (lines 14-15): Information on sensitivity and uncertainty values would be quite useful.

We added these infos in the experimental section

  • Page 6 - Figure 3: The bioluminescence using the concentration of 35 μgL-1 Hg2+and 50 μgL-1 Hg2+ has not been easily differentiable. Could the authors give more information about this issue, please?

Data are taken from samples prepared in the same way (same starting OD, environmental and growth conditions) except for the mercury concentration, as described in the Material and Methods section. The samples underwent an incubation process evolving in independent ways. Thus their evolution is not fully predictable and small differences can be observed as in our case. Our goal was to show, with the optical measurements, the trend as a function of the mercury content.

  • Page 7 – Figure 4: The average bioluminescence signals have presented unexpected behavior to concentration below 1 μgL-1 Hg2+with the increase of incubation time. Could the authors give more information about this issue, please?

The points below 1 μg/L are very close to the optical system sensitivity limit, hence we do not have signal to acquire useful information on their behaviour. However, we reported a work of Berset et al. (Ref. 52 now in the manuscript) describing an induction model developed for an arsenic reporter construct that could be useful to better understand the intracellular mechanisms involved in the metal sensing and that can be assumed for mercury due to the similar genetic recombination developed. We clarified this point on page 7 of the manuscript.

Reviewer 3 Report

Dear editor,

In this paper, the authors propose an advanced and miniaturized method for the specific and highly sensitive detection and quantification of divalent mercury in water combining the properties of both whole-cell biosensor and silicon-based technology. I have some issue as follows.

  1. How did authors detect or avoid the mutual interference between different concentration of mercury standard solutions?
  2. The surrounding temperature is a significant factor. How to evaluate it?

Author Response

We would like to thank all the referees for the suggestions that help us to improve the quality of our manuscript. In the following we report the referees’ comments/suggestions in bold and our answers below each one in italic.

  1. How did authors detect or avoid the mutual interference between different concentration of mercury standard solutions?

Each acquisition chamber was optically insulated, using a black tape. We verified there is no optical cross-talk during the measurements. The test was made by measuring a heavily contaminated solution and leaving the other chamber empty (control). No signal was detected in the control chamber. We added a sentence on page 4 in the text to clarify this point.

  1. The surrounding temperature is a significant factor. How to evaluate it?

We placed our system into a polystyrene box to minimize temperature variations during the measurements and the lab temperature is set to 24°C by the conditioning system. Nevertheless we monitored, the humidity and temperature by a temperature and Humidity data logger (mini TH). Temperature is stable within 1°C and humidity does not change during the measurement time.